# The Precious Few Grams of Glucose During Exercise

**DOI:** 10.3390/ijms21165733

**Published:** 2020-08-10

**Authors:** George A. Brooks

**Affiliations:** Exercise Physiology Laboratory, University of California, Berkeley, 5101 VLSB, Berkeley, CA 94720-3140, USA; gbrooks@berkeley.edu

**Keywords:** glucose, metabolism, exercise, homeostasis, euglycemia, lactate

## Abstract

As exercise intensity exceeds 65% of maximal oxygen uptake carbohydrate energy sources predominate. However, relative to the meager 4–5 g blood glucose pool size in a postabsorptive individual (0.9–1.0 g·L^−1^ × 5 L blood = 18–20 kcal), carbohydrate (CHO) oxidation rates of 20 kcal·min^−1^ can be sustained in a healthy and fit person for one hour, if not longer, all the while euglycemia is maintained. While glucose rate of appearance (i.e., production, Ra) from splanchnic sources in a postabsorptive person can rise 2–3 fold during exercise, working muscle and adipose tissue glucose uptake must be restricted while other energy substrates such as glycogen, lactate, and fatty acids are mobilized and utilized. If not for the use of alternative energy substrates hypoglycemia would occur in less than a minute during hard exercise because blood glucose disposal rate (Rd) could easily exceed glucose production (Ra) from hepatic glycogenolysis and gluconeogenesis. The goal of this paper is to present and discuss the integration of physiological, neuroendocrine, circulatory, and biochemical mechanisms necessary for maintenance of euglycemia during sustained hard physical exercise.

## 1. Introduction

Maintenance of blood glucose concentration in a narrow range is a major physiological priority [1]. Because of its essential role in brain metabolism, an acute fall in glucose results in disorientation, seizure, and even death. On the other hand, a chronic elevation in blood glucose concentration, results in glucose toxicity, diabetes, and metabolic syndrome. The problem of maintaining blood glucose homeostasis (euglycemia) is always a major challenge, particularly after high carbohydrate containing meals and during physical exercise when the demand for carbohydrate (CHO) energy use is huge compared to blood glucose content. In this manuscript CHO use pertains to the major body energy sources that are glycogen, lactate, and glucose. In contrast, the term refers to the hexose in the blood. Given its importance in health and disease the matter of glucoregulation in exercising individuals has received much interest [2,3,4,5,6]. As exercise intensity exceeds 65% of maximal oxygen uptake carbohydrate energy sources predominate [7]. However, relative to the meager teaspoon (4 to 5 g) of glucose in the blood of a 70 kg postabsorptive individual (0.9–1.0 g·L^−1^ blood × 5 L blood = 18–20 kcal) [6,8], carbohydrate oxidation rates of 20 kcal·min^−1^ can be sustained in healthy physically fit males [9,10,11] and females [12,13] while euglycemia is maintained. Noteworthy is that while the glucose rate of appearance (i.e., production, Ra) from splanchnic sources in a postabsorptive person can rise 2–3 fold during exercise, working and non-working muscle and adipose tissue glucose use must be restricted. Without those restrictions hypoglycemia would result in less than one minute during hard exercise because blood glucose disposal rate (Rd) could easily exceed glucose production (Ra) from hepatic glycogenolysis and gluconeogenesis (GNG).

## 2. Glucose Blood Pool Size and Energy Content

Blood glucose concentration remains constant for exercise durations of one hour or more in untrained subjects, while in trained subjects blood (glucose) may actually rise slightly during hard exercise such as at the lactate threshold (LT). In young, healthy and reasonably fit individuals blood (glucose), and therefore blood glucose pool size, is well regulated in moderate intensity exercise of reasonable duration (Figure 1). The relationship between blood glucose pool size and total body energy expenditure can be estimated at least in two ways, from indirect calorimetry or isotopic dilution plus indirect calorimetry. Using standard approximations blood volume of a 70 kg postabsorptive individual approximates 5 L (L). In the same postabsorptive person blood glucose concentration approximates 100 mg·dL^−1^, or 1 g·L^−1^ making the blood glucose pool size 5 g (=1 g·L^−1^ blood × 5 L blood). From the enthalpy of glucose (4 kcal·g^−1^), the energy content of blood glucose in a standard sized person approximates 20 kcal total (=5 g blood glucose × 4 kcal·g^−1^). The 20 kcal energy equivalent of the blood glucose pool compares to the resting, postabsorptive energy rate of approximately 1.2 kcal·min^−1^ (=0.25 L·min^−1^ O_2_/min × 4.8 kcal L·min^−1^ O_2_ consumed) of which 40% may be CHO oxidation. Assuming that muscle glycogen oxidation is quiescent in the resting postabsorptive person then the glucose oxidation rate per minute approximates 0.48 g·min^−1^ or almost 10% of the available blood glucose pool.

Studies utilizing indirect calorimetry give an appreciation of the range of CHO oxidation rates in the transition from rest to exercise and yield important information on the change in energy substrate partitioning from a mix of CHO and lipid at rest to near complete CHO dependence during hard exercise [7,11,14]. In healthy young physically active individuals capable of sustaining an oxygen uptake of 3 L·min^−1^ CHO oxidation rates of 10–15 kcal·min^−1^ can be sustained for one hour, if not longer, without a decline in blood glucose concentration [9,11] (Figure 1). This observation pertains to both young healthy and physically active males [10,11] and females [12,13]. In the example shown, young men were studied at rest and at 45 and 65% of VO_2_max before and after 3 months of supervised endurance training (Figure 2) [14].

In combination, indirect calorimetry and glucose tracer methodology can be used to parse the contributions of glucose, other CHO energy sources and lipids to total energy production. In Figure 2 the third exercise column representing CHO oxidation after training, a decrease in CHO oxidation is seen compared to the second exercise column representing the same exercise power output that elicited 65% of VO_2_max prior to training. In all exercise conditions glucose provided only 10–20% of total energy expenditure, while oxidation of other CHO energy sources (i.e., glycogen and lactate) provided 70–80% of total energy expenditure. More impressive is that highly trained male athletes capable of sustaining oxygen uptake rates of 4–5 L·min^−1^ exhibit carbohydrate oxidation rates of 20–25 kcal·min^−1^ that can be sustained for hours [15]. What is remarkable in such individuals is that even considering mild expansion of blood volume, it still approximates 5 L, with a corresponding blood glucose pool size of 5 g. The point of these mathematical manipulations is that blood glucose pool size and energy contents are modest in comparison to use demands put on them that range from a few percent to close to 100% percent each minute. Maintaining euglycemia during prolonged and hard physical exercise is certainly a major physiological challenge. How does the body do it? By a combination of endocrine and cardiovascular responses, lactate shuttles and glucose shunts the body maintains euglycemia.

## 3. Does Muscle Glucose Use Rise During Exercise?

The answer to the question, does muscle glucose use rise during exercise, seems obvious; however, the answer may be “yes” or “no” depending on one’s viewpoint, particularly if glucose use is evaluated in absolute or relative terms. In absolute terms the answer is “yes” because splanchnic glucose production rises several fold during exercise (Figure 3) [10], and because blood flow is redirected (shunted) to working muscle and away from non-working muscle and other tissue beds (Figure 4) [10,16]. Hence, it is possible to state that glucose is shunted to working muscle during exercise. In Figure 3 note in the third exercise column, a decrease in glucose flux after training is seen compared to the second exercise column representing the power output that elicited 65% of VO_2_max prior to training. However, blood glucose flux was greatest at 65% of VO_2_max after training.

In juxtaposition to the idea that glucose use is increased during exercise and after endurance training one can compare blood glucose flux and working muscle glucose uptake and oxidation (Figure 3 and Figure 4) to the gain in muscle power output and CHO oxidation during exercise (10–20 fold) (Figure 2). From this perspective, the 2–3 fold rise in glucose flux (Figure 3) and working muscle glucose uptake and oxidation (Figure 4) are relatively small by comparison to the use of other CHO and lipid energy sources.

Another way to assess whether glucose use increases during exercise and after endurance training is to examine fractional glucose uptake in resting and exercising muscle (Figure 5). Even though vasodilation in working muscle beds increases the surface area available for glucose uptake, at the working muscle level, fractional extraction of glucose coursing through working muscle remains at about 5% during rest and exercise, both before and after endurance training. Increased muscle perfusion during exercise compared to rest, and after training (Figure 6) mean that glucose extraction from arterial blood must occur in less capillary transit time thus posing a potential limit to glucose fractional extraction (Figure 5).

## 4. Counter Regulatory, Feed Forward, and Feedback Responses

Maintenance of euglycemia is typically, and profoundly understood in terms of the actions of insulin and its counter regulatory hormones [6]. However, much like the control of breathing during exercise when hyperpnea occurs without a fall in arterial O_2_ or a rise in CO_2_, conditions that cause significant, but small increases in resting pulmonary minute ventilation, maintenance of euglycemia during exercise and recovery can easily be misinterpreted to mean that there are no error or other signals to suppress insulin secretion or provoke increases in glucagon, cortisol, or the catecholamine secretion. Nonetheless, a fall in circulating insulin during exercise is considered a major advantage as working muscle has little need for insulin to stimulate glucose uptake [17,18] while non-working muscle and adipose develop a transitory “glucose resistance” from the fall in insulin, thus preserving more available glucose to be shunted to working muscle [10].

When evaluating the roles of insulin and the counter-regulatory hormones during exercise it is helpful to consider the role of exercise intensity and training status as mediated through the sympathetic nervous system. In general, exercise training tends to lower hormone levels in resting individuals and dampen the responses to exercise stress, an exception being that male athletes demonstrate heightened catecholamine responses to maximal efforts [1]. As shown in Figure 7, endurance training lowers circulating insulin and glucagon levels in resting individuals [10]. Also as shown in the Figure 7, and typical for exercise insulin falls while euglycemia is maintained. While insulin falls during exercise, circulating glucagon remains unchanged or falls except in untrained persons during hard exercise when glucagon rises, again in absence of a change in blood (glucose). This makes the purported role of the insulin/glucagon ratio (I/G) during difficult exercise to construe, in part because glucagon released from the pancreas first passes the liver.

Results of studies on exercising dogs involving hepatic portal vein catheterization clearly show a role for glucagon in regulating hepatic glucose production [19]. In studies in exercising humans, in who attempts at hepatic portal vein catheterization would be unethical, the role of glucagon in supporting euglycemia can be seen during prolonged, moderate intensity exercise particularly if blood (glucose) declines [20]. However, as emphasized previously [6], hepatic metabolism of glucagon masks its role. Fortunately, the role of glucagon was revealed in studies on exercising men when somatostatin was given to block glucagon secretion and hypoglycemia resulted [20].

Results of longitudinal training studies such as illustrated in Figure 7 point to an extremely important role of the sympathetic nervous system that exerts profound effects on the cardiovascular system during exercise. Hence, changes in circulating insulin, glucagon and the I/G when euglycemia is maintained during exercise are likely attributable to the effects of catecholamines norepinephrine and epinephrine that respond exponentially to gradations in exercise intensity whether expressed in absolute or relative terms (Figure 8) [21]. Indeed, in a study on male athletes engaged in hard exercise, Kjaer and colleagues observed feed-forward sympathetic responses that resulted in very high epinephrine that stimulated hepatic glycogenolysis to an extent that hyperglycemia resulted and was characterized as “inaccurate glucoregulation” [22]. In subsequent experiments in which sympathetic responses were elicited by the addition of arm to leg exercise the same investigators repeated the observation of an epinephrine response that raised blood glucose and suppressed insulin during exercise [23].

The intricate balance of feed-forward and counter regulatory responses that regulate glycemia during exercise likely also possesses components of feedback control beyond those understood as part of counter regulation and the push and pull on blood glucose concentration. One example of feedback control is the influence of lactate on catecholamine levels. Observed during the course of lactate-clamp studies of resting and exercising men [24,25] was the effect of lactate infusion on suppressing circulating levels of norepinephrine and epinephrine (Figure 9) [26]. Endogenously produced [22] or exogenously administered epinephrine [27,28] has the effect of stimulating glycogenolysis and raising circulating lactate levels, that in turn, suppress catecholamine levels.

## 5. Glucose Regulation by Direct and Indirect Feedback—Something Was Missing

Not completely satisfied with the proposition that glucoregulatory hormone responses during exercise can explain observed phenomena, others have asked the question, paraphrased “How does the liver know what the muscles are doing?” In our experience we have studied the effects of nutrition, exercise intensity, and exercise training, gender, menstrual cycle phase, oral contraceptives, aging, and high altitude exposure; hence, we and many others have observed that euglycemia is well maintained over a broad range of challenges. We wondered how does the liver and associated organs get it right? In an exhaustive, highly innovative, and immensely precise set of experiments B.K. Pedersen and colleagues have addressed the issue of how the liver and adipose knows what muscles are doing by showing that human muscle is a secretory organ that produces cytokine IL-6 [29,30,31,32]. Based on the experience with macromolecules such as glucose and oxygen some would have thought it would be impossible to measure arteriovenous difference (a–v) of IL-6 differences across resting and working muscle beds because it is in such low abundances in blood, but the group accomplished the task [32]. Similarly, some would not have imagined it possible for the group to measure (a–v) differences across the hepatosplanchnic bed; yet again they accomplished the task [29]. Moreover Pedersen and colleagues have studied intracellular signaling at both ends of the muscle-liver and muscle-adipose axes which includes a major role for lactate signaling in IL-6 release from muscle [30]. Hence, via IL-6 Pedersen and colleagues showed how muscle could communicate with liver during exercise allowing for glucoregulation without the typical error signal of a falling blood glucose level that would act to suppress insulin and raise insulin antagonist hormones. Moreover, why should IL-6 be the only myokine involved in glucoregulation? Time and further research will tell if there are others.

From a broad physiological perspective, why were we slow to recognize the roles of direct and indirect feedback from muscle in physiological regulation? Typically, sympathetic system activation precedes physical activity and provides feed-forward, anticipatory regulation of numerous cardiovascular, pulmonary, endocrine, and other responses. Importantly, regardless of the mode or form, physical activity is activated from that cerebral motor cortex. Then, immediately as part of the Metaboreflex muscles communicate directly via Types III and IV afferents to brain cardiovascular [33], and likely other regulatory centers. By simultaneously releasing macro- [34] and micro- molecules [31] via the vasculature muscles communicate with multiple organ systems including the brain [35], liver [36], pancreas [37], and adipose [38]. This all happens before blood glucose, insulin or glucagon levels change during exercise, if they change at all.

## 6. Glucose, Glycogen, and Lactate Interactions: The Lactate Shuttle and Glycemia

As noted above, the blood glucose pool and splanchnic glucose production are too small to support CHO energy needs during exercise; hence glycogen [39] and lactate fulfill those essential roles [34]. Because of dietary and circadian rhythms, and because of difficulty in measuring hepatic glycogen content [40], body muscle and liver glycogen contents vary, but values of 280 and 480 g of liver and muscle glycogen in fed individuals are reasonable [41]. After an overnight fast, when hepatic and muscle glycogen reserves have been depleted to maintain euglycemia during sleep, values of 100 and 400 g of liver and muscle glycogen contents are reasonable [6]. As emphasized above, these values not only serve to illustrate the importance of glycogen reserves for providing energy (Figure 2), but the values also serve to support the importance of gluconeogenesis and show the role of the Lactate Shuttle in maintaining euglycemia by providing an alternative oxidizable substrate, but also in supporting gluconeogenesis [11,36,42,43]. Hence, while 75–80% of lactate formed during exercise is disposed of within working red muscle and heart [44,45,46], the remainder is disposed of via gluconeogenesis [36,43,47]. In the example shown (Figure 10) gluconeogenesis was determined by means of primed-continuous infusions of D2-Glucose and [3-^13^C]Lactate and the incorporation ^13^C from infused lactate into plasma glucose. Exercise and exercise training increase GNG from lactate; the effect is demonstrable even at rest. In other experiments the effects of exercise training and gender on GNG were determined using the principle of reversible and irreversible tracers with primed-continuous infusions of D2-Glucose (irreversible tracer) and [1-^13^C]Glucose (reversible tracer) [12,14]. In this case the lesser flux rate with the reversible (^13^C-tracer) compared to the dideuterated (D-tracer) is assumed to reflect recycling of the ^13^C-tracer through gluconeogenesis, i.e., the Cori Cycle. Because similar results were obtained using the two tracer methodologies it is safe to conclude that lactate is by far the most important gluconeogenic precursor during postabsorptive exercise.

## 7. A Teaspoon of Goodness: Brain Glucose and Lactate Interactions

Exercise and other physiologists can easily associate metabolism with muscle, nerve cardio-pulmonary, circulatory, and endocrine systems, and even the integument when thermoregulation is an issue. Seemingly meaning of the terms “neuromuscular” and “neuromuscular activity” are sometimes lost. However, except for simple reflexes, the brain is ultimately the controller of breathing, essential aspects of cardiovascular control, and physical activity mode, intensity, and duration. Likewise, the brain is responsible for attaching meaning, perception, expression, and emotions to physical activities. Consequently, the term “exercise” ultimately denotes “the brain”.

Because of its anatomical location, importance, and delicacy, historically there have been few attempts to measure brain metabolism. Seminal attempts [41,49,50] showed a high relative rate of brain metabolism that was dependent on glucose as a fuel. More recent results support the role of cerebral glucose metabolism in health, during exercise and after trauma [51,52], but recent results also show that, depending on availability, lactate can represent an important brain energy source. Indeed, in comparing brain glucose and lactate metabolism, some would argue that lactate is neither an alternative or supplementary brain energy source, but rather that lactate is the brain’s preferred energy fuel, with the job of astrocytes being to glycolyze and provide neurons with their essential energy source, lactate [53,54]. For studies on humans in vivo, the importance of cerebral CHO metabolism cannot be simply estimated by net balance measurements alone (i.e., ((a–v)Glucose) (Brain Blood Flow)) vs. ((a–v)Lactate) (BBF)), but rather by the combination of (a–v) balance and carbon tracer uptake and excretion as ^13^CO_2_ [34,35]. For example, in comatose traumatic brain injury (TBI) patients, the role of lactate in brain fueling was found to be impressive because most (70–80%) of circulating blood glucose was produced via gluconeogenesis from lactate [47,51]. As well, net lactate uptake directly provided 12% of brain fuel. Hence, directly (12%), and indirectly via gluconeogenesis (45%), most (57%) of brain fuel was from lactate even if most brain energy was provided by glucose (Figure 11) [55]. However, for purposes of discussion in this paper suffice it to assert that because of its constancy and relatively high concentration (4–5 mM vs. 0.5–1 mM, and twice the molecular mass (180 vs. 89 g·mol^−1^), glucose is preferred over lactate (or ketones) as brain fuel in resting, postabsorptive individuals. For example, at plasma (lactate) of 1 mM on a net basis lactate contributes 12% of the brain’s total energy requirement [51,56] while glucose contributes 82%. It is for the reason of fueling the healthy brain that the RDA for CHO Nutrition was established at 130 g·day^−1^ [55].

## 8. Glycemia and Nutrition: Splanchnic and Hypothalamic Interactions

It goes without saying that short- and long-term blood glucose regulation depends on nutrition that is ultimately dependent on appetite. Appetite regulation is a complicated and active area of research [57,58], suffice it to state that in the arcuate nucleus of the hypothalamus effects of the appetite stimulating (Feeding) (Neuropeptide Y (NPY)) Center are balanced against those of the appetite inhibitory (Satiation) (Pro-opio-melanocortin (POMC)) Center. Hormones that inform the hypothalamic centers that regulate appetite include insulin, ghrelin, leptin, peptide YY (PYY), glucagon like peptide-1 (GLP-1), and others. Results of classic studies of Mayer and colleagues [59], as well as practical experience, indicate that exercise has both appetite-inhibiting as well as appetite-stimulating effects. For instance, after competitive 400, 800, and 1500 m races no one is hungry; appetite is suppressed or hours and returns later. Anecdotally, after a series of Lactate Clamp studies involving exogenous Na^+^-lactate infusion and exercise after a 12 h fast [24,25,60], male study participants did not complain of hunger. Regrettably, the short-term effect of hard exercise on mechanisms of exercise suppression on appetite has been little investigated, but effects of acidosis, hyperthermia and changes in blood metabolite levels can be imagined. Fortunately, however, there is growing data on the suppressive effect of lactate on appetite [61,62]. In this regard recent results of Hazell and colleagues in studying endocrine responses to hard exercise support the hypothesis that lactatemia results in appetite suppression via ghrelin secretion [63,64], ghrelin released from the stomach upon the appearance of food being one of the major stimulators of appetite via action on the arcuate nucleus in the hypothalamus.

The lactatemia of exercise may have peripheral as well as central effects on the hypothalamus. It has been recognized that microbiota in the gastrointestinal (GI) tract produce lactate (3 carbons) as well as short chain fatty acids (SCFA) such as butyrate (4 carbons) [34]. The ghrelin receptor (growth hormone secretagogue receptor (GHSR)-1α) is a G-protein-coupled receptor expressed throughout both the stomach and GI tract. In a recent report [65] investigators have found that SCFAs, lactate, and other bacterial excretions in the GI tract are able to attenuate ghrelin-mediated signaling through the *GHSR-1*. Hence, in combination with lactate produced by gut microbiota, the high blood lactate of exercise can enter the bowel via sodium-mediated monocarboxylate transporters (sMCT) and attenuate ghrelin receptor signaling showing yet another novel route of communication between the gut microbiota and the host, and vice versa.

Returning to the theme of exercise effects on glucose metabolism, in hard exercise circulating levels of appetite stimulators insulin, and ghrelin would be suppressed by epinephrine, lactate, and redistribution of blood flow. Those appetite-suppressing effects of lactate would persist into recovery and may be part of the beneficial effects of high intensity interval training (HIIT) on metabolic regulation [66].

## 9. Glycemia and Hydration in Exercise: Discoveries that Led to Founding of an Industry

As a fuel energy source glucose can be provided in several ways, from: the blood pool, glycogenolysis, gluconeogenesis, and nutrition. Because the focus of this article is on maintenance of euglycemia during exercise, and in view of the associated needs for providing fluid, electrolytes, and energy in activities lasting more than one hour sports drinks contain carbohydrates. As explained above, the blood glucose pool is miniscule compared to the CHO energy needs in hard and prolonged training and exercise. Body glycogen stores are much larger, but they too can be exhausted during prolonged, vigorous exercise bouts [39]. Thus, the need to provide fluid, electrolytes, and an energy source, or sources can be helpful for maintaining euglycemia and prolonged (≥1 h) high exercise power outputs.

Solutions containing 4–6% glucose can be effective in providing 1 g·min^−1^ [67]. This uptake (60 g·h^−1^, 240 kcal·h^−1^) dwarfs blood glucose availability. More concentrated glucose solutions are sometimes less efficacious as gastric emptying is limited and gastrointestinal distress results during exercise. However, by adding the disaccharide sucrose (glucose + fructose), fructose and lactate polymers investigators are able to deliver 1.5 g·min^−1^ of CHO energy [68,69]. As part of electrolyte replacement sports drinks include sodium ions that have the benefit of taking advantage of sodium-mediated solute transport including the sGLUT transporter for glucose and the sMCT transporter for lactate [34]. As well, conversion of fructose to lactate, probably in intestinal mucosa, hastens nutrient delivery [69].

## 10. Who’s the Big Boss? Autonomic Responses or Insulin?

The present paper is based more on human subject research and places greater emphasis on roles of the sympathetic nervous system, myokines, and lactate for maintaining euglycemia during exercise than do previous treatments of the same subject [6]. However, the role of insulin is not to be minimized. In humans glycemia and glucose metabolism cannot be maintained, and life cannot long continue without insulin. In an untreated Type-1 (insulin deficient) person the combined the counter-regulatory actions of insulin antagonists (catecholamines, glucagon, cortisol, gut hormones) and myokines are insufficient in maintaining glycemia and support life. Insulin is essential always, not just in the postprandial period after eating, but during exercise also. As described above and by others [70], insulin withdrawal in human exercise is a permissive mechanism of supporting glycemia by allowing the actions of counter regulatory hormones and other mechanisms in the face of huge energy demands. By analogy, the role of insulin can be likened to that of a symphony conductor. To uneducated eyes and ears the conductor’s role may appear more obvious in playing of a thunderous crescendo piece, such as the fourth movement of Beethoven’s 5th Symphony, than his or her role in conducting a gentle and contemplative piece such as Debussy’s Prelude to the Afternoon of a Faun. By analogy, the conductor’s role is exactly the same in directing both musical pieces and is like that of insulin in metabolic regulation. Whether in soft diminuendo, as in exercise, or at full throttle after a large, CHO-containing meal, in the regulation of glycemia insulin is very much controlling.

## 11. Summary

The presence and availability of only a few grams of glucose, one of many energy substrates in blood, is essential for normal physiology. Deviations from a small range of blood glucose concentrations can have both short- and long-term effects on CNS functioning and progression of metabolic diseases. The relatively small amount of blood sugar in the face of a huge range of metabolic demands requires fine-tuning and constant coordination of a variety of different strategies to maintain the normative range. These strategies involve producing more glucose via hepatic glycogenolysis and gluconeogenesis, consuming dietary CHO, using less glucose, diverting (shunting) the available glucose supply to where it is most needed (working muscle and heart) while at the same time limiting glucose uptake by other insulin responsive tissues (inactive muscle and adipose), using alternative endogenous fuel energy sources (fatty acids and lactate), and consuming a diet providing alternative fuel energy sources. These strategies are orchestrated by both behavior in terms of the timing and composition of dietary foods and beverages, and by physiology. The neuroendocrine responses involve both feed-forward and feedback components. For exercise complicated set of endocrine responses involving the sympathetic nervous system and its capabilities for direct and endocrine signaling initiate cardiopulmonary, cardiovascular, and other fight and flight autonomic responses via norepinephrine and epinephrine that cardiovascular control serve to direct the limited glucose supply where it is needed (heart, brain, working skeletal muscle), and shunt blood flow and glucose delivery away from other tissues (Figure 12). Sympathetic responses also initiate pancreatic β-cell inhibition and diminution of insulin secretion by epinephrine, pancreatic α-cell stimulation of glucagon secretion by epinephrine and declining blood (glucose), if that occurs in prolonged exercise. The liver is very sensitive to glucagon, but hepatic metabolism partially masks the role of glucagon on regulation of glycemia during exercise. Feedback responses include those mediated by the Metaboreflex and actions of IL-6 and possibly other myokines as well as lactate secreted by working muscle. In aggregate, these mechanisms all serve to maintaining glycemia in a narrow range.

## Figures and Tables

**Figure 1 ijms-21-05733-f001:**
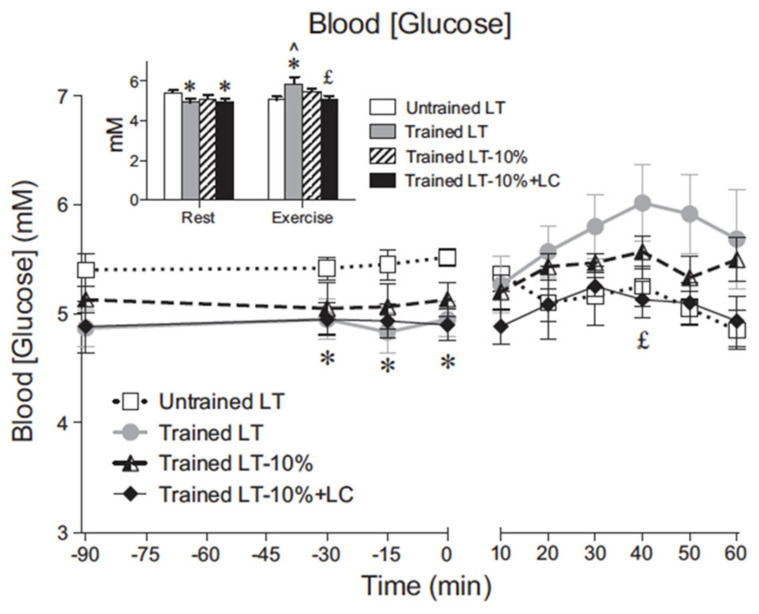
Blood glucose concentration across time in resting and exercising young men. Values are means ± SE; *n* = 6 for untrained and trained groups. The inset shows steady-state rest and exercise glucose concentrations for each condition. LT, lactate threshold; LT-10%, 10% below the LT workload; LT-10% LC, 10% below the LT workload with a lactate clamp. ^ Significantly different from rest within condition (*p* < 0.05). * Significantly different from untrained (*p* < 0.05). ^£^ Significantly different from trained LT (*p* < 0.05). Note that even in the young healthy cohort regular physical exercise (training) significantly decreases resting, 12 h, overnight fasted blood (glucose). Note also that in trained men blood (glucose) increased during hard exercise at the lactate threshold (LT), but that exogenous lactate infusion (lactate clamp, LC) decreases blood (glucose). From [11].

**Figure 2 ijms-21-05733-f002:**
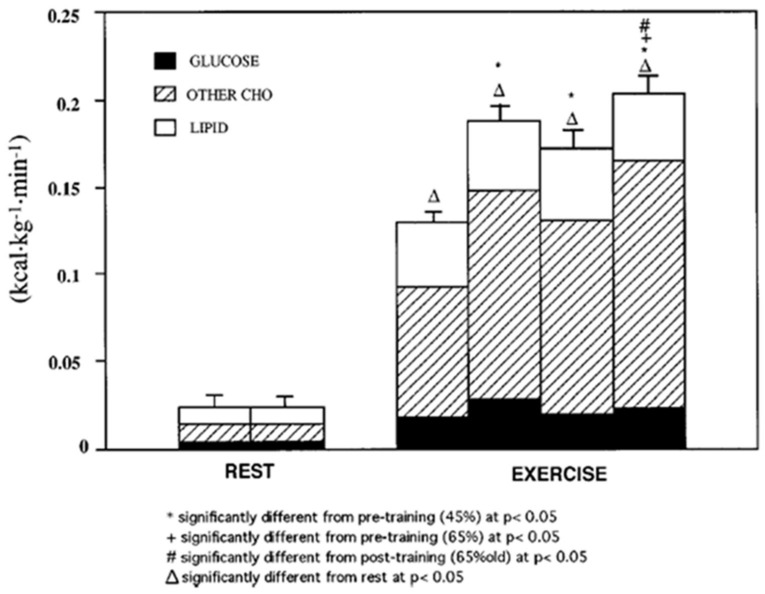
Energy generated from oxidation of carbohydrate (CHO) and lipid sources. Subjects were studied at 45 and 65% of peak rate of oxygen consumption (VO_2_peak) before 9 week of endurance training (45% pre and 65% pre, respectively) and then after training at same absolute power output that elicited 65% VO_2_peak before training (65% old (absolute, ABT)) and same relative power output that elicited 65% VO_2_peak after training (65% new (relative, RLT)). Error bars, SE for total energy expenditure only; *n* = 10 subjects. Note that compared to rest, energy expenditure during exercise increases more than 10 times. Note also that at rest glucose oxidation represents approximately half of total CHO oxidation that represents only 25% of energy expenditure. During exercise glucose oxidation represents only 10–20% of total energy expenditure and approximately 25% of total CHO oxidation. From [14].

**Figure 3 ijms-21-05733-f003:**
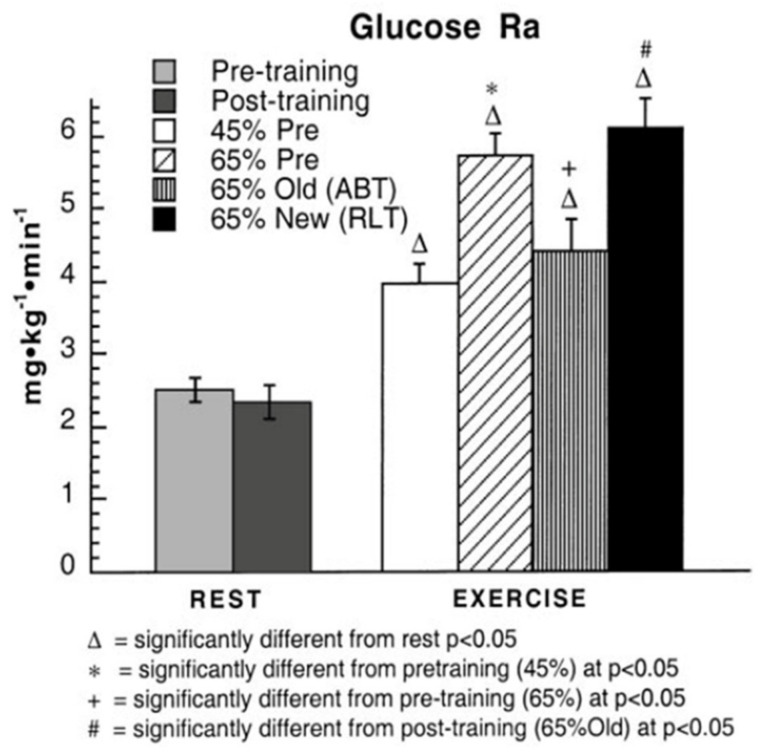
Blood glucose appearance rates (Ra), or flux in healthy young men during rest and exercise, before and after training. Subjects were studied at 45 and 65% of peak rate of oxygen consumption (VO_2_peak) before 9 week of endurance training (45% pre and 65% pre, respectively) and then after training at same absolute power output that elicited 65% VO_2_peak before training (65% old (absolute, ABT)) and same relative power output that elicited 65% VO_2_peak after training (65% new (relative, RLT)). Values are means ± SE; *n* = 8–9. Note that compared to rest, exercise increases tracer-measured blood glucose rate of appearance 2–3 times. Comparing the second and third columns on the right in which men exercised at the same exercise power output before and after training, glucose Ra and Rd (not shown) decreased with training. However, the greatest glucose Ra was observed during hard, 65% VO_2_max exercise after training (last column). From [10].

**Figure 4 ijms-21-05733-f004:**
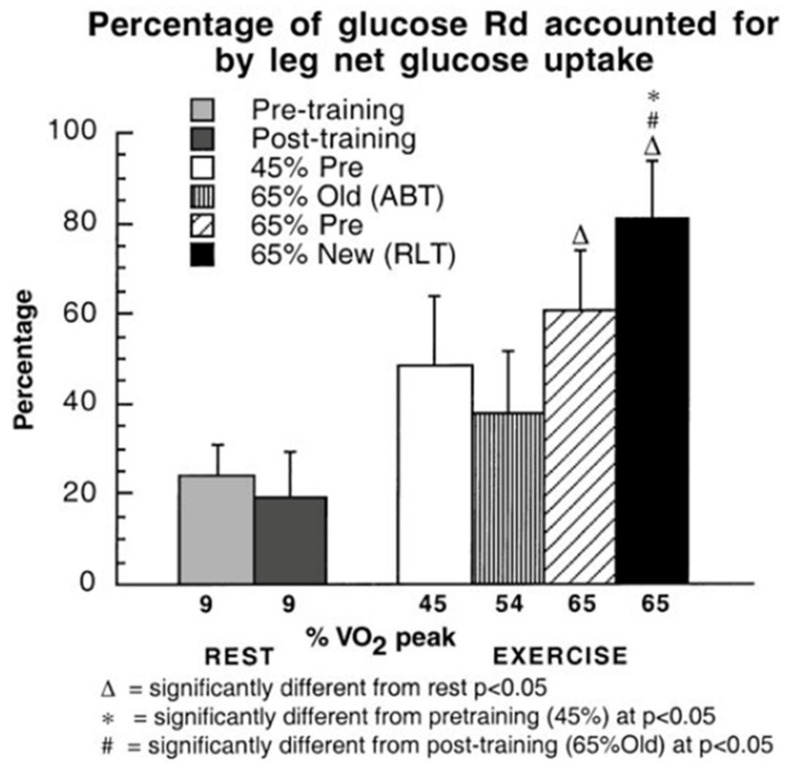
Percentages of glucose Rd accounted for by leg net glucose uptake as a function of relative exercise intensity during rest and exercise in healthy young men, before and after training. Subjects were studied at 45 and 65% of peak rate of oxygen consumption (VO_2_peak) before 9 week of endurance training (45% pre and 65% pre, respectively) and then after training at same absolute power output that elicited 65% VO_2_peak before training (65% old (absolute, ABT)) and same relative power output that elicited 65% VO_2_peak after training (65% new (relative, RLT)). Values are means of last 30 min of exercise for glucose Rd and 1 h of exercise for net glucose uptake ± SE; *n* = 6–8, same subjects as in Figure 3. Note the relatively large increase in leg muscle glucose uptake during exercise, especially after training when leg power output increased approximately 25% over pre-training exercise intensity. While this figure shows a relative shunt of available blood glucose to working muscle, as shown in Figure 2 the energy role of blood glucose to the overall metabolic response to exercise is small. From [10].

**Figure 5 ijms-21-05733-f005:**
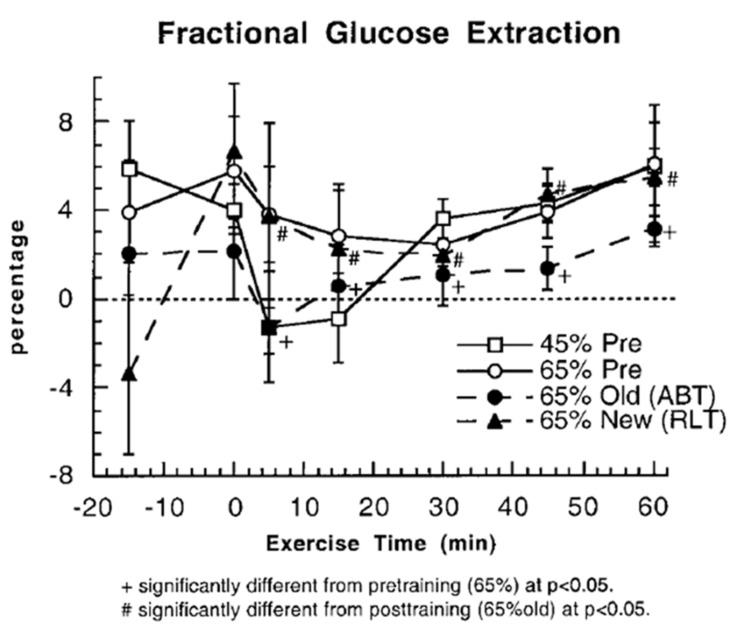
Fractional glucose extractions (%) determined from arterial-venous concentration differences (a–v) in resting and exercising young men, before and after endurance training. Subjects were studied at 45 and 65% of peak rate of oxygen consumption (VO_2_peak) before 9 week of endurance training (45% pre and 65% pre, respectively) and then after training at same absolute power output that elicited 65% VO_2_peak before training (65% old (absolute, ABT)) and same relative power output that elicited 65% VO_2_peak after training (65% new (relative, RLT)). Values are means ± SE; *n* = 6–8, same subjects as in Figure 3 and Figure 4, but show variability due to variances around measuring arterial and venous blood glucose values. There does not appear to be a training-induced increase in resting or working muscle glucose fractional extraction. From [10].

**Figure 6 ijms-21-05733-f006:**
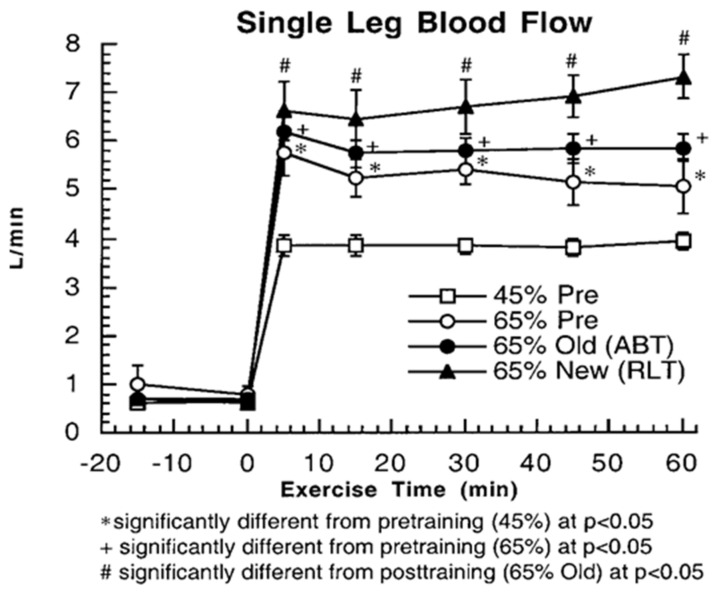
Single leg blood flow rates in resting and exercising healthy young men from thermodilution at rest and during exercise, before and after training. Subjects were studied at 45 and 65% of peak rate of oxygen consumption (VO_2_peak) before 9 week of endurance training (45% pre and 65% pre, respectively) and then after training at same absolute power output that elicited 65% VO_2_peak before training (65% old (absolute, ABT)) and same relative power output that elicited 65% VO_2_peak after training (65% new (relative, RLT)). Values are means ± SE; *n* = 6–8, same individuals and treatments as in Figure 3, Figure 4 and Figure 5. Exercise and exercise training increase leg muscle blood flow compared to rest and during the untrained condition. However, there does not appear to be a training-induced increase in resting or working muscle glucose fractional extraction (Figure 5). From [10].

**Figure 7 ijms-21-05733-f007:**
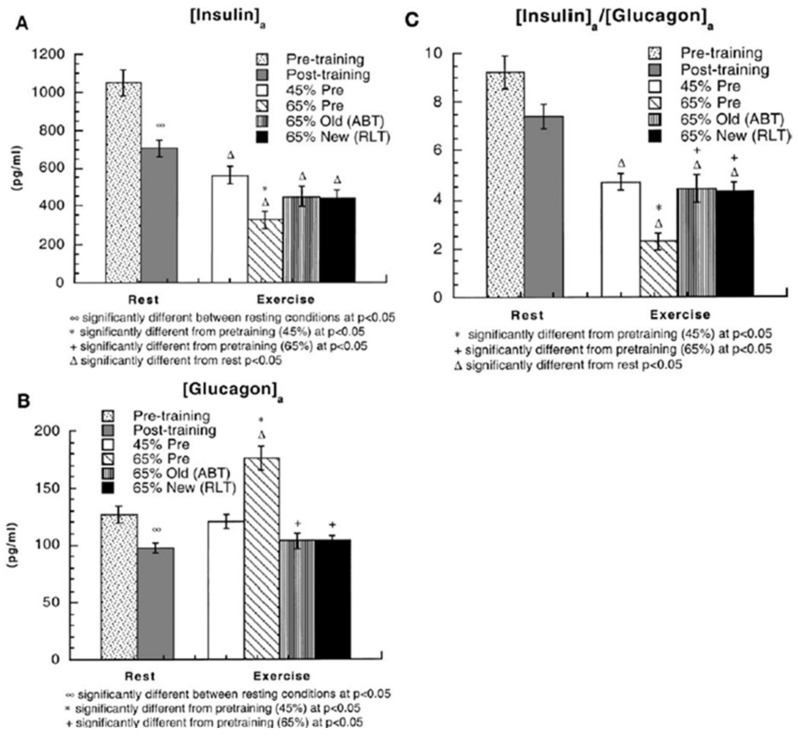
(**A**) arterial insulin concentrations ((insulin)_a_) in healthy young men during rest and exercise, before and after training. Subjects (same as in Figure 3, Figure 4, Figure 5 and Figure 6) were studied at 45 and 65% of peak rate of oxygen consumption (VO_2_peak) before 9 week of endurance training (45% pre and 65% pre, respectively) and then after training at same absolute power output that elicited 65% VO_2_peak before training (65% old (absolute, ABT)) and same relative power output that elicited 65% VO_2_peak after training (65% new (relative, RLT)). Values are means ± SE; *n* = 8–9. (**B**) Arterial glucagon concentrations ((glucagon)_a_) during rest and exercise, before and after training. (**C**) Arterial insulin-to-glucagon ratios ((insulin)_a_/(glucagon)_a_) during rest and exercise, before and after training. Endurance training has the effect of decreasing insulin and glucagon levels and the I/G. Insulin levels declined during exercise; glucagon rose in the hard 65% VO_2_peak condition before training, but otherwise remained unchanged or declined slightly during exercise. See text for discussion. From [10].

**Figure 8 ijms-21-05733-f008:**
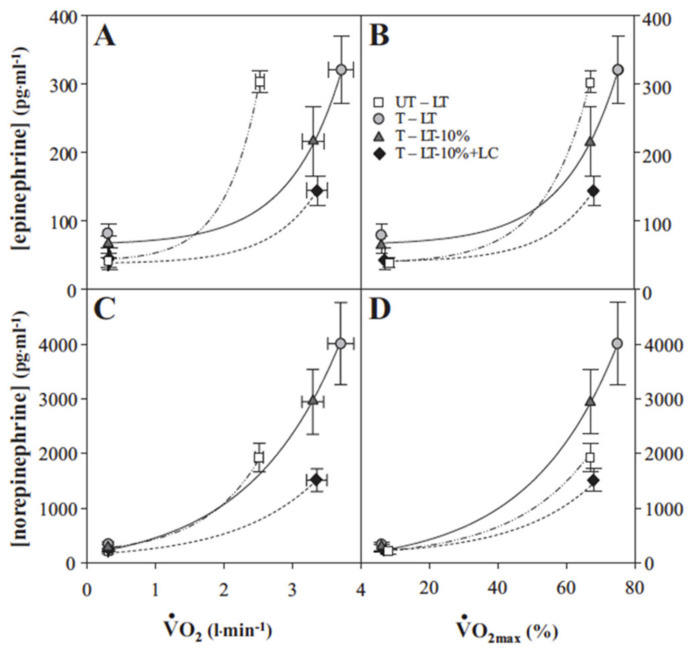
Plasma epinephrine (**A**,**B**) and norepinephrine (**C**,**D**) as functions exercise intensity as given by absolute (VO_2_) (**A**,**C**) and relative (to % VO_2_peak) (**B**,**D**). Untrained subjects (UT) were studied at the exercise power output that elicited the lactate threshold (LT). Trained subjects (T) studied at power outputs that elicited the LT, 10% below the LT (LT-10%), and LT-10% plus exogenous lactate infusion raising blood (lactate) (i.e., lactate clamp, LC) to that eliciting the LT (LT-10% + LC). Metabolic rates elicited at rest and exercise in the present and previous studies involving subjects with different physical fitness status. VO_2_max values (means ± SEM) of UT and T subjects are 3.7 ± 0.1 and 5.0 ± 0.3 L·min^−1^, respectively. Whether expressed as absolute or relative values both trained and untrained subjects respond to increments in exercise intensity with exponential increments in circulating catecholamines. From [21].

**Figure 9 ijms-21-05733-f009:**
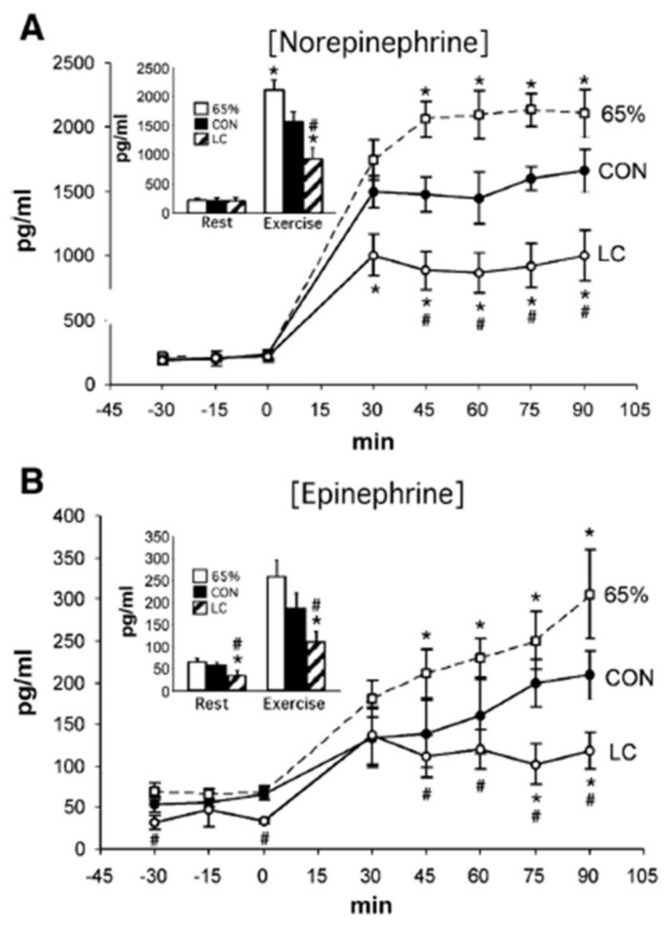
Plasma norepinephrine (**A**) and epinephrine (**B**) concentrations across time during rest and exercise. Shown are data for control (CON) and lactate clamp (LC) at 55% VO_2_peak; 65% indicates 65% VO_2_peak. Insets are mean rest and exercise for each condition. *p* < 0.05 is significantly different from CON (*) and significantly different from 65% (^#^). Results show a suppression of plasma catecholamine levels by exogenous lactate. See text for discussion. From [26].

**Figure 10 ijms-21-05733-f010:**
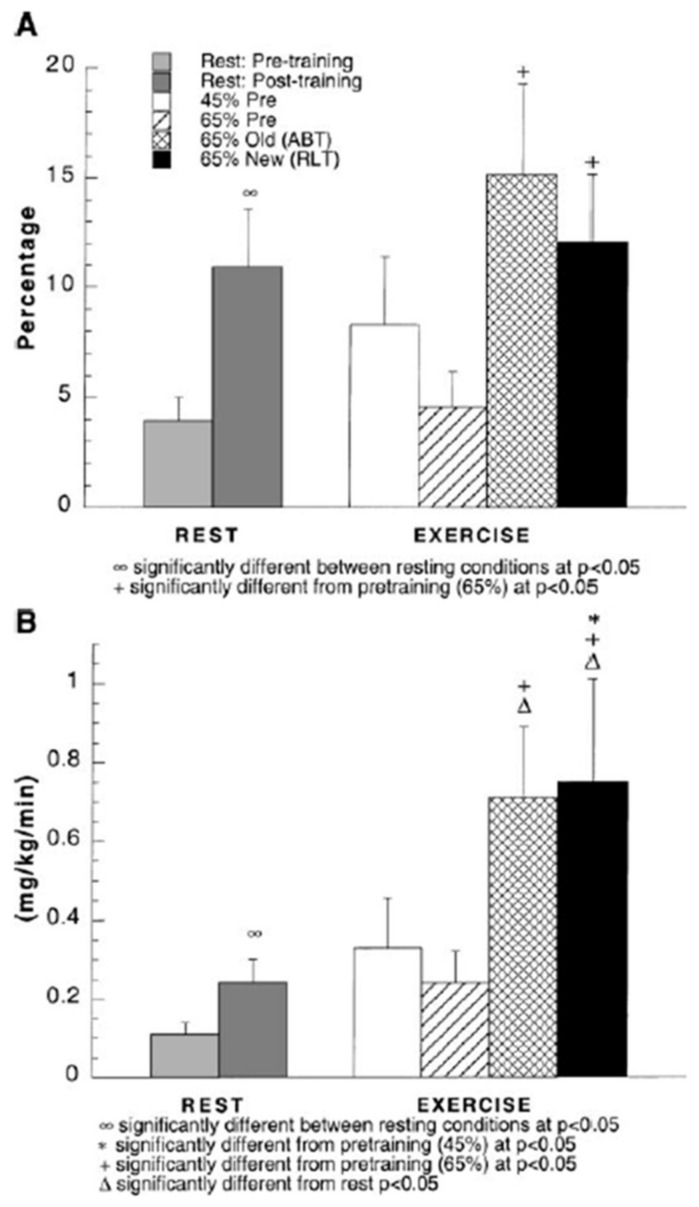
(**A**) Percent glucose appearance rate (Ra) from gluconeogenesis (GNG) during rest and exercise, before and after training. (**B**) Estimated GNG during rest and exercise, before and after training. Subjects were studied at 45 and 65% of peak rate of oxygen consumption (VO_2_peak) before 9 week of endurance training (45% pre and 65% pre, respectively) and then after training at same absolute power output that elicited 65% VO_2_peak before training (65% old (absolute, ABT)) and same relative power output that elicited 65% VO_2_peak after training (65% new (relative, RLT)). Values are means ± SE; *n* = 8–9. Although most lactate formed during steady rate exercise is disposed of within working skeletal muscle and elsewhere in the body such as heart [44,46,48], gluconeogenesis (GNG) accounts for 20–25% of lactate disposal during exercise. Hence, GNG is a major avenue of lactate disposal and the most important GNG precursor during exercise. From [36].

**Figure 11 ijms-21-05733-f011:**
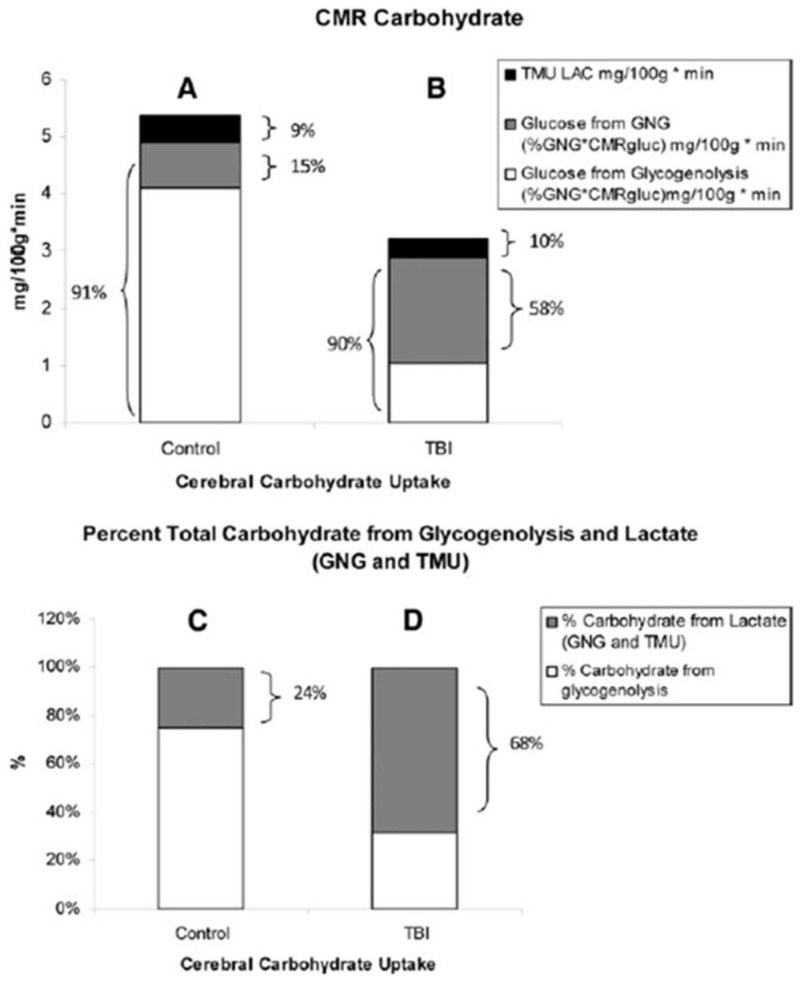
Histograms of the absolute and relative contributions to total cerebral carbohydrate (CHO) from lactate, glucose from gluconeogenesis (GNG), and glucose from hepatic glycogenolysis in healthy control subjects (left, panels **A**,**C**), and traumatic brain injury (TBI) patients (right, panels **B**,**D**). Compared with panel (**A**,**B**) shows the decrease in cerebral metabolic rate for glucose (CMRgluc) following TBI, but also shows increased contributions of lactate and glucose from GNG to total cerebral CHO uptake after TBI. A comparison of panels (**C**) (control) and (**D**) (TBI) shows the large increase in percentage cerebral CHO uptake contributed by lactate, directly, or indirectly from GNG, following TBI. These figures demonstrate the direct and indirect means by which lactate and gluconeogenesis from lactate support the healthy as well as injured brain. From [47].

**Figure 12 ijms-21-05733-f012:**
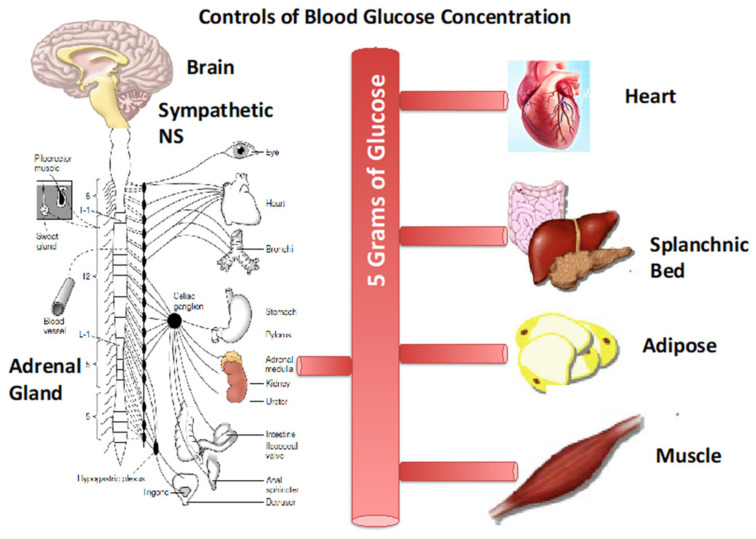
A mere 5 g, a teaspoon, of blood sugar (glucose) is regulated by a coordinated set of neuroendocrine responses involving both feed-forward and feedback components. For exercise the responses involve the sympathetic nervous system (SNS) and its capabilities via post-ganglionic nerves that release norepinephrine and the adrenal medulla that releases epinephrine for direct and endocrine signaling that initiate cardiopulmonary, cardiovascular, and other fight and flight autonomic responses. Cardiovascular control serves to direct the limited glucose supply where it is needed (heart, brain, working skeletal muscle), and shunt blood flow and glucose delivery away from other tissues. In the splanchnic bed the pancreas secretes insulin (β-cells), glucagon (α-cells), and somatostatin (*δ*-cells). As well, gastric secretion of ghrelin, GLP-1, and PYY affect appetite and eating behavior that provides major support of glycemia. Feedback control includes the Metaboreflex, secretion of myokines (e.g., IL-6) and lactate from working muscle, and changes in blood (glucose), as well as epinephrine from the adrenal medulla that inhibits pancreatic secretions. In adipose lipolysis is inhibited by lactate via c-AMP and CREB as well as TGF-β2 released from liver under the influence of circulating lactate. See text for details. Figure modified from [1] and other sources including [6].

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
