# Peer review of "The Precious Few Grams of Glucose During Exercise"

_ijms, 2020, doi:10.3390/ijms21165733_

Round 1
Reviewer 1 Report
Overall this is a well written and interesting review article discussing the various tissues and physiological processes involved in the control of blood glucose levels during exercise.
Major comments:
1) It would be helpful if the authors could rephrase the final sentence in the abstract into a purpose or goal statement for the review.
2) The analogy comparing insulin to a symphony conductor in section 10 requires knowledge of the musical works and conductor to follow, and this may be further complicated by readers whose native language is not English. Although the spirit of the analogy is appreciated, it would enhance the overall clarity of the review if it was simplified or removed.
Author Response
Response to the Reviewers ijms-876425R0
Reviewer’s Comments in Ariel font, Author’s response in Italicized Times font.
Reviewer #1
Overall this is a well written and interesting review article discussing the various tissues and physiological processes involved in the control of blood glucose levels during exercise.
Major comments:
1) It would be helpful if the authors could rephrase the final sentence in the abstract into a purpose or goal statement for the review.
Thank you for the constructive comment; the text has been edited as suggested.
2) The analogy comparing insulin to a symphony conductor in section 10 requires knowledge of the musical works and conductor to follow, and this may be further complicated by readers whose native language is not English. Although the spirit of the analogy is appreciated, it would enhance the overall clarity of the review if it was simplified or removed.
Good point that may apply to native English speakers as well. Text has been edited for clarity for those of different cultures and languages.
Again, thank you for the constructive comments.
Reviewer 2 Report
This review did a good job explaining the maintenance of euglycemia during exercise in a concise and understandable way. Inclusion and explanation of past research done on this topic were exquisite. The following are minor suggestions I would like the author to consider:
The terms glucose and carbohydrate are used interchangeably throughout the manuscript. Please add a sentence in the introduction indicating that these terms are used interchangeably and do not mean different things.
Throughout the manuscript kcal/min and L/min are used please change these to kcal·min-1 and L·min-1 to be consistent with your reporting of other units.
Abstract
Page 1, line 7: comma after “uptake”
Page 1, line 9: (0.9-1.0 g L x 5 L blood 18-20 kcal) should be changed to (0.9-1.0 g·L-1 x 5 L blood x 4 kcals/g CHO = 18-20 kcals)
Page 1, line 10: change “an hour” to “one hour”
Page 1, line 10: add a comma after “longer”
Page 1, line 14: add a comma after “substrates”
Page 1, line 15: change “a minute” to “one minute”
Introduction
Page 1, line 31: see comment for line 9
Page 1, line 33: Please re-write this sentence so that it does not start with “And”
Page 1, line 35: Please re-write this sentence so that it does not start with “Otherwise”
Glucose Blood Pool Size and Energy Content
Page 2, line 15: add a comma after “min”
Page 2, line 15: change “an hour” to “one hour”
Page 2, line 19: delete extra period at the end of the sentence
Page 3, line 8: add “o” to “t”
Page 3, line 27: Add an s to and a comma at the end of “shunt”
Does Muscle Glucose Use Rise During Exercise?
Page 5, line 12: “too” should be changed to “to”
Counter Regulatory, Feed Forward, and Feedback Responses
Page 7, line 13: Is partial pressure of O2 and CO2 being referred to here? Please be more detailed.
Page 7, line 27: add a comma after “exercise”
Page 7, line 28: add a comma after “exercise”
Page 8, line 15: change “who” to “which”
Page 8, line 15: add a comma after “unethical”
Page 8, line 18: add “in” between the words “revealed” and “studies”
Glucose Regulation by Direct and Indirect Feedback – Something was Missing
Page 10, line 19: define arteriovenous difference before using the acronym
Page 10, line 20: change the comma after “abundance” to a semicolon
Page 11, line 3: add a comma after “IL-6”
Page 11, line 12: change “that” to “the”
Page 11, line 14: delete the second “by” in the second sentence
Page 11, line 15: add a comma after “vasculature”
Glucose, Glycogen and Lactate Interactions: The Lactate Shuttle and Glycemia
Page 11, line 30: add “is” after “remainder”
Page 11, line 38: add a hyphen between 13C and tracer to maintain consistency
A Teaspoon of Goodness: Brain Glucose and Lactate Interactions
Page 14, figure 11: define TMU in the caption
Glycemia and Nutrition: Splanchnic and Hypothalamic Interactions
Page 14, line 20: add “f” in front of “or”
Page 15, line 3: The use of “suppressive” and “suppression” in this sentence seems redundant
Page 15, line 10: add “(SCFA)” after “short chain fatty acids”
Page 15, line 11: delete “both”
Page 15, line 19: Leptin is included in the list of appetite stimulators. I believe leptin is an appetite inhibitor.
Glycemia and Hydration in Exercise: Discoveries that led to Founding of an Industry
Page 15, lines 25-27: The sentence, “Because the focus of this article is on maintenance of euglycemia during exercise, and because of the associated needs for providing fluid, electrolytes and energy in activities lasting more than an hour sports drinks contain carbohydrates.” does not flow well. Please consider revising.
Page 15, line 38: “Sodium ion” should be changed to “sodium ions” and “has” to “have”
Who’s the Big Boss? Autonomic Responses or Insulin?
Page 15, lines 45-47: The sentence: “However, the role of insulin is not to be minimized because in humans glycemia and glucose metabolism cannot be maintained, and life cannot continue despite actions of insulin antagonists and myokines.” Sounds a bit confusing, please re-write.
Summary
Page 16, line 24: check the number of spaces between “pancreatic-cell”
Page 17, line 2: add a comma after “exercise”
Page 17, line 8: add β and α
Page 17, line 12: add a comma after “adipose”
Page 17, line 13: add beta-2 after “TGF”
Author Response
Response to the Reviewers ijms-876425R0
Reviewer’s Comments in Ariel font, Author’s response in Italicized Times font.
Reviewer #2
This review did a good job explaining the maintenance of euglycemia during exercise in a concise and understandable way. Inclusion and explanation of past research done on this topic were exquisite. The following are minor suggestions I would like the author to consider:
Thank you for the authoritative and helpful review. However, with Palatino and Times fonts we will encounter problems with super and subscripts.
The terms glucose and carbohydrate are used interchangeably throughout the manuscript. Please add a sentence in the introduction indicating that these terms are used interchangeably and do not mean different things.
Thank you for the helpful comment; the terms were not intended to be used interchangeably so clarity on the point is important. Hence, brief definitions have been added even if they disrupt text flow slightly.
Throughout the manuscript kcal/min and L/min are used please change these to kcal·min-1 and L·min-1 to be consistent with your reporting of other units.
Units have been edited to be consistent within the manuscript.
Abstract
Page 1, line 7: comma after “uptake”
Sorry, can’t find the place after editing.
Page 1, line 9: (0.9-1.0 g L x 5 L blood 18-20 kcal) should be changed to (0.9-1.0 g·L-1 x 5 L blood x 4 kcals/g CHO = 18-20 kcals)
OK, g·L-1, but not kcals, just kcal.l.
Page 1, line 10: change “an hour” to “one hour”
OK, but unnecessary.
Page 1, line 10: add a comma after “longer”
OK, but it was there.
Page 1, line 14: add a comma after “substrates”
Suggestion is appreciated, but declined. The text is pithy and too many parenthetical phrases disturb text flow.
Page 1, line 15: change “a minute” to “one minute”
OK.
Introduction
Page 1, line 31: see comment for line 9
OK, but, again, with Palatino and Times fonts we can expect problems with super (e.g., g·L-1 ) and subscripts (e.g., Oxygen, O2). Please check the units to make sure that bub and superscript s are legible (e.g., Oxygen, O2, g·L-1,kcal·min-1), here expressed in 12 pitch Arial font. Note the superior ability of the presentation.
Page 1, line 33: Please re-write this sentence so that it does not start with “And.”
Changed, but unnecessarily so. The text was fine for common usage.
Page 1, line 35: Please re-write this sentence so that it does not start with “Otherwise”
OK.
Glucose Blood Pool Size and Energy Content
Page 2, line 15: add a comma after “min”
Sorry, can’t find this.
Page 2, line 15: change “an hour” to “one hour”
OK, could also use “per.”
Page 2, line 19: delete extra period at the end of the sentence
Sorry, can’t find this, but you have permission to change.
Page 3, line 8: add “o” to “t”
Sorry, can’t find this, but you have permission to change.
Page 3, line 27: Add an s to and a comma at the end of “shunt”
OK.
Does Muscle Glucose Use Rise During Exercise?
Page 5, line 12: “too” should be changed to “to”
Counter Regulatory, Feed Forward, and Feedback Responses
Page 7, line 13: Is partial pressure of O2 and CO2 being referred to here? Please be more detailed.
Page 7, line 27: add a comma after “exercise”
Sorry, can’t find this, but you have permission to change.
Page 7, line 28: add a comma after “exercise”
Sorry, can’t find this, but you have permission to change.
Page 8, line 15: change “who” to “which”
CORRECT AS IS; PLEASE DO NOT CHANGE.
Page 8, line 15: add a comma after “unethical”
OK.
Page 8, line 18: add “in” between the words “revealed” and “studies”
OK.
Glucose Regulation by Direct and Indirect Feedback – Something was Missing
Page 10, line 19: define arteriovenous difference before using the acronym
OK.
Page 10, line 20: change the comma after “abundance” to a semicolon
Other changes, but Not this.
Page 11, line 3: add a comma after “IL-6”
Sorry, can’t find this, no permission to change.
Page 11, line 12: change “that” to “the”
Sorry, can’t find this, no permission to change.
Page 11, line 14: delete the second “by” in the second sentence
OK.
Page 11, line 15: add a comma after “vasculature”
OK.
Glucose, Glycogen and Lactate Interactions: The Lactate Shuttle and Glycemia
Page 11, line 30: add “is” after “remainder”
OK.
Page 11, line 38: add a hyphen between 13C and tracer to maintain consistency
A Teaspoon of Goodness: Brain Glucose and Lactate Interactions
Page 14, figure 11: define TMU in the caption
Glycemia and Nutrition: Splanchnic and Hypothalamic Interactions
Page 14, line 20: add “f” in front of “or”
Page 15, line 3: The use of “suppressive” and “suppression” in this sentence seems redundant
Page 15, line 10: add “(SCFA)” after “short chain fatty acids”
OK.
Page 15, line 11: delete “both”
No, add “stomach.”
Page 15, line 19: Leptin is included in the list of appetite stimulators. I believe leptin is an appetite inhibitor.
Excellent!!! Thank you.
Glycemia and Hydration in Exercise: Discoveries that led to Founding of an Industry
Page 15, lines 25-27: The sentence, “Because the focus of this article is on maintenance of euglycemia during exercise, and because of the associated needs for providing fluid, electrolytes and energy in activities lasting more than an hour sports drinks contain carbohydrates.” does not flow well. Please consider revising.
Revised.
Page 15, line 38: “Sodium ion” should be changed to “sodium ions” and “has” to “have”
Revised.
Who’s the Big Boss? Autonomic Responses or Insulin?
Page 15, lines 45-47: The sentence: “However, the role of insulin is not to be minimized because in humans glycemia and glucose metabolism cannot be maintained, and life cannot continue despite actions of insulin antagonists and myokines.” Sounds a bit confusing, please re-write.
Good points. The section has been revised.
Summary
Page 16, line 24: check the number of spaces between “pancreatic-cell”
OK.
Page 17, line 2: add a comma after “exercise”
Sorry, can’t find this, but you have permission to change.
Page 17, line 8: add β and α
OK.
Page 17, line 12: add a comma after “adipose”
Sorry, can’t find this, but you have permission to change.
Page 17, line 13: add beta-2 after “TGF”
OK
One again, thank you for the excellent and thorough review. Perhaps machine learning has been used in the process. As we see, machine analysis has its advantages and disadvantages. Thanks especially for the comments on vagaries in science.